# Newborn Screening for Severe Combined Immunodeficiency: Lessons Learned from Screening and Follow-Up of the Preterm Newborn Population

**DOI:** 10.3390/ijns9040068

**Published:** 2023-12-15

**Authors:** Amy Gaviglio, Michael Lasarev, Ruthanne Sheller, Sikha Singh, Mei Baker

**Affiliations:** 1Association of Public Health Laboratories, Silver Spring, MD 20910, USA; amy.gaviglio@aphl.org (A.G.);; 2Department of Biostatistics and Medical Informatics, School of Medicine and Public Health, University of Wisconsin-Madison, Madison, WI 53726, USA; lasarev@biostat.wisc.edu; 3Wisconsin State Laboratory of Hygiene, School of Medicine and Public Health, University of Wisconsin-Madison, Madison, WI 53726, USA; mei.baker@slh.wisc.edu; 4Department of Pediatrics, School of Medicine and Public Health, University of Wisconsin-Madison, Madison, WI 53726, USA

**Keywords:** severe combined immunodeficiency, newborn screening, follow-up, preterm newborn

## Abstract

Newborn screening (NBS) for Severe Combined Immunodeficiency (SCID) by measurement of T-cell receptor excision circles (TRECs) successfully identifies newborns with SCID and severe T-cell lymphopenia, as intended. At the same time, NBS programs face the challenge of false positive results, with a disproportionately high number in the premature newborn population. This study evaluates TREC values and SCID screening outcomes in premature newborns and elucidates evidence-based SCID screening practices that reduce unnecessary follow-up activities in this population. De-identified individual SCID newborn screening data and aggregate SCID screening data were obtained from seven states across the US for babies born between 2018 and 2020. Relevant statistics were performed on data pooled from these states to quantify screening performance metrics and clinical impact on various birth and gestational age categories of newborns. The data were normalized using multiples-of-the-median (MoM) values to allow for the aggregation of data across states. The aggregation of NBS data across a range of NBS programs highlighted the trajectory of TREC values over time, both between and within newborns, and provides evidence for improved SCID screening recommendations in the premature and low birth weight population.

## 1. Introduction

Severe Combined Immunodeficiency (SCID) represents a group of monogenic diseases characterized by profound T-cell lymphopenia (TCL) with variable defects in B-cell lymphocytes. Owing to the lack of a functioning immune system, individuals with SCID experience high mortality rates if left untreated. Immune function can be restored with treatment, typically hematopoietic stem cell transplantation (HSCT), or in some cases, gene therapy or enzyme replacement therapy [1].

Given that early treatment (within the first few months of life) results in significantly improved outcomes and survival rates [2], SCID was determined to be an appropriate candidate condition for newborn screening (NBS) and, in 2010, was added to the United States’ Recommended Uniform Screening Panel (RUSP). Newborn screening for SCID is primarily accomplished through the measurement of T-cell receptor excision circles (TRECs). TRECs are non-replicating small circles of deoxyribonucleic acid (DNA) that are formed during T-cell receptor gene rearrangement within the thymus, and thus, serve as a surrogate biomarker of thymic function. While SCID is the primary target of NBS, measurement of TRECs also identifies other conditions with T-cell lymphopenia (TCL), including syndrome disorders of T-cell development [3], disorders of thymic stromal cell development, idiopathic T-cell lymphopenia, and secondary T-cell lymphopenia due to factors like preterm birth and very low birth weight [4].

NBS programs typically employ either real-time polymerase chain reaction (RT-PCR) or an end-point PCR method to screen for SCID/TCL. Screening results may be reported in one of three ways: (1) quantitative TREC copy numbers, (2) cycle threshold (Ct) values, or (3) multiple of the median (MoM), which is the ratio of the result of an individual measurement (TREC copy number or Ct value) to the median result for measurements in the appropriate population.

Over time, the universal implementation of SCID newborn screening in the United States has revealed some challenges in the interpretation of TREC screening values, including a relatively high number of false positive results, especially a disproportionally high number in the preterm newborn population [5,6]. Generally, these false positive results are due to the relative immaturity of a preterm infant’s immune system and T-cell development at the time of newborn screening. As a result of the high number of screen-positive results in this population, preterm newborns are often screened multiple times or undergo unnecessary diagnostic testing in efforts to delineate or resolve the screening results. NBS programs throughout the country have expressed interest in how best to perform SCID screening and follow-up in this population, to reduce clinical burden while maintaining high-quality performance metrics.

Growing awareness that screening in the preterm population presents particular challenges in the interpretation of low TREC values, has resulted in many NBS programs implementing special recommendations and procedures for screen-positive results in this group. Generally, these have involved recommending continuous repeat screenings until the low TREC values are resolved or the child reaches full-term (i.e., 37 weeks corrected gestational age) [7]. However, despite these altered recommendations, and because of increased survivability in earlier and earlier gestational ages, the preterm population remains an enigma. As such, further assessment and evidence-based recommendations are needed to improve the quality of screening for these babies.

This study was designed to better understand TREC values and SCID screening outcomes in preterm newborns, through a multi-state collaboration with systematic data collection and analysis. This report assesses the association between TREC copy numbers and gestational age and/or birth weight, which, in turn, provides an explanation for the disproportionally higher screening false positive rate in premature infants, and the rationale for the need for special consideration in this population.

## 2. Materials and Methods

### 2.1. Data Collection

The Association of Public Health Laboratories (APHL) newborn screening and genetics program performed outreach to its members—specifically all newborn screening programs across the US—to request data to contribute to this study. Data were shared via a secure portal over the span of several months, and ultimately, there were three distinct data sets collected from seven participating states. 

Screen-Negative Preterm Cohort Data Set: The first data set focused on preterm infants born between 2018 and 2020 with screen-negative results and an initial screening specimen that was collected between 24 and 48 h of age. Additional exclusion and inclusion criteria can be found below. Participating states provided a de-identified list of all cases that met these criteria, along with each newborn’s birthweight, reported biological sex, gestational age, and birth year.

From these lists, a random sample was selected for each state and additional data were requested, including age at specimen collection and TREC screening values. 

Screen-Positive Preterm Cohort Data Set: The second data set focused on preterm infants born between 2018 and 2020 with screen-positive results. Data requested for the screen-positive preterm cohort included birthweight, gestational age, age at specimen collection, screening values, transfusion status, follow-up recommendations, final outcomes, and flow cytometry data, if available. This information was requested for all samples associated with a newborn who had a screen-positive result, regardless of if other specimens were subsequently screen-negative.

Aggregate Data Set: The third data set focused on aggregate data across 2018, 2019, and 2020. Participating states provided the total number of babies screened, the total number of babies with out-of-range results for SCID/TCL, the total number of babies with out-of-range SCID/TCL results where a repeat screen was requested, and the total number of babies with out-of-range SCID/TCL results where clinical evaluation was recommended for each of the calendar years. Final outcomes for out-of-range results were also provided.

Data Collection and Curation: To support states with querying the requested data, several technical assistance calls were held, and a robust data dictionary (Appendix A) was provided. Upon submission, data were reviewed for adherence to exclusion/inclusion criteria, data elements, and data formatting. Any questionable data were reviewed between the state program and the study team for final determination.

Exclusion Criteria: Newborns with a birth weight greater than 2500 g or a gestational age greater than or equal to 37 weeks were excluded from this study. Additionally, newborns with unsatisfactory specimens (including, but not limited to, poor quality specimens, transit issues, no deoxyribonucleic acid (DNA) amplification, or any other reason a laboratory may classify a specimen as unsatisfactory or invalid) were excluded. Newborns with screen-positive results and other disorders on the NBS panel were not included.

Birth Weight versus Gestational Age: Though prematurity is determined by gestational age, birth weight is sometimes used as a proxy by NBS programs (instead of trying to collect gestational age which may be measured in several ways) as it is a more clearly defined measurement and, thus, an easier data element to collect. In cases where TREC results are shown by birth weight, note that this measure is being used as a proxy for gestational age. 

### 2.2. Statistical Analysis

Negative Cohort: For each participating state that collected gestational age, a random sample of newborns was selected at each gestational age in weekly increments between 23 and 36 weeks. Sample size (at each week of gestation) was set to 20% of each state’s total for a given week of gestational age; if the 20% rule resulted in five or fewer newborns, then the total number available at that gestational age was selected. States that collected birthweight instead of gestational age were sampled using the same 20% rule based on 14 equally spaced intervals of weight between 400 and 2500 g. Multiples of the median (MoM) for each specimen from the negative cohort were computed by normalizing TREC or Ct values by the median value given by each state. Quantile regression [8] was used to determine whether MoM was associated with gestational age, which entered the model as a 3-knot restricted cubic spline to allow for both linear and non-linear patterns. Models were separately fit for instances where MoM was based on TREC, rather than Ct. In cases where birth weight was provided instead of gestational age, the quantile regression model for MoM used a 6-knot restricted cubic spline for birth weight. A supplemental analysis treated birth weight as an ordered factor with three levels: extremely low birth weight (ELBW; <1000 g), very low birth weight (VLBW; <1500 g), and low birth weight (LBW; <2500 g). Jonckheere’s test [9] was used to explore whether MoM steadily decreased with increasing birth weight classification. 

Positive Cohort: The percentage of samples having each type of final diagnosis (normal repeat screen, true positive, or false positive) was summarized for the positive cohort according to gestational age, which was coarsened into seven separate intervals (each spanning two weeks) between 22 and 36 weeks of gestation. Logistic regression was used to estimate the fraction of true positive outcomes as a function of gestational age, to better understand at what age the true positive fraction most rapidly increased. A generalized estimating equation (GEE) [10] was used to explore how the marginal probability that a state’s testing algorithm would classify a repeat sample as normalized (i.e., negative) versus non-normalized (borderline or positive) changed as a function of gestational age. These probabilities were estimated for a series of potential thresholds ranging between 25 and 36 corrected weeks of gestation.

Aggregate Results: The Positive Predictive Value (PPV) was calculated for a selection of programs that provided the most comprehensive aggregate data. The overall PPV for normal weight/full-term babies was compared to the calculated PPV for low birth weight/premature babies.

## 3. Results

Three states provided data based on TREC values, while four states supplied data based on Ct; one of these four states only collected birth weight and was not able to provide gestational age. For this state, birth weight was used as a proxy for gestational age. Gestational age ranged from 23 to 36 weeks for those states that recorded this information. 

Median MoM at 23 weeks gestation for TREC-based measures (*n* = 11,302) was 0.32 (95% CI: 0.26–0.37) and increased to 0.92 (95% CI: 0.90–0.93) by 36 weeks, an estimated 2.9-fold increase over the 13-week span (95% CI: 2.5–3.5 fold; *p* < 0.001) (Figure 1).

The pattern was inverted for Ct-based measures (*n* = 10,125), which demonstrated a decrease in MoM with increasing gestational age. Initially (at 23 weeks gestational age), the median MoM was 1.034 (95% CI: 1.027–1.042) and decreased 2.6% (95% CI: 1.9–3.4%; *p* < 0.001) over 13 weeks to a median MoM of 1.007 (95% CI: 1.006–1.008) at 36 weeks gestational age (Figure 2).

Ct-based MoM was also found to decrease as a function of birth weight (Figure 3), based on one state that recorded this information rather than gestational age. Birth weight among *n* = 13,339 newborns ranged from 409 to 2500 g and could also be classified as extremely low birth weight (ELBW < 1000 g; *n* = 273, 2.05%), low birth weight (VLBW < 1500 g; *n* = 937, 7.02%), or low birth weight (LBW < 2500 g; *n* = 12,129, 90.92%). The data showed a convincing (*p* < 0.001) decrease in median MoM with increasing birth weight; each 500 g increase in birth weight was associated with a 0.96% reduction in the median MoM (95% CI: 0.88–1.03% reduction). Figure 4 shows the same data organized according to the three categories of birthweight. Although the groups showed a strong trend of decreasing MoM with increasing weight (*p* < 0.001; Jonckheere’s test), individual comparisons among the three groups (without regard to the ordering of categories) found no real difference between ELBW and VLBW (*p* = 0.201; Steel-Dwass multiple comparison procedure) while each of the two lower groups both differed from the LBW group (*p* < 0.001 for each; Steel-Dwass).

The final outcome (normal repeat screen on a second sample [NRS], false positive [FP], or true positive [TP]) from 1943 separate screen-positive cohort infants was cross-classified according to gestational age, which was categorized into seven separate 2-week-long intervals covering the range from 22 to 36 weeks. False positive outcomes were either ‘false positive’ or ‘transient T-cell lymphopenia’, while true positive outcomes consisted of either ‘true positive’ or ‘non-SCID T-cell lymphopenia’. The case definitions developed by NewSTEPs for these terms were used for the purpose of this study [11]. The prevalence of each type of outcome is given in Table 1 and shown graphically in Figure 5. Tests revealed that the average true positive fraction increased by 0.81 (95% CI: 0.42–1.20, *p* < 0.001) percentage points per 2 weeks of gestational age. Logistic regression revealed that the prevalence of true positive screens increased rapidly after a gestational age of ~30 weeks and stayed relatively constant (at a TP fraction of ~2%) prior to the 30-week landmark (Figure 6). 

The calculated PPV for True/Leaky SCID and non-SCID T-Cell lymphopenias in the LBW/Premature population was 0.5% and 7.3%, respectively, for those programs providing complete aggregate data. Alternatively, the PPV for True/Leaky SCID and non-SCID T-cell lymphopenias in full-term babies was 7.7% and 38.5%, respectively, in those same programs.

The Interpretation made using a state’s screening algorithm, either screen-negative or screen-borderline/positive, was analyzed for association with corrected gestational age, which was calculated as the infant’s recorded gestational age plus age (in weeks) at the time of sample collection. In total, 7042 tests were performed on 2124 infants from the six states that recorded gestational age. The number of specimens collected per infant ranged from 1 to 13, with 63.3% of infants contributing at most three specimens, and 16.7% contributing five or more specimens (see Appendix A for the full distribution). Estimates based on the GEE model reveal that the likelihood that a repeat sample has normalized stays constant near 65% if collected at or before ~32 weeks corrected gestational age, but then gradually increases to 76% (95% CI: 71–80%) if the sample is collected at or after 36 weeks corrected gestational age (Figure 7).

## 4. Discussion

Owing to the known challenges in screening for SCID/TCL in preterm newborns, NBS programs have utilized a number of tactics to try to reduce the impact on this population, largely through the use of gestational age or birth weight-based reference ranges or targeted serial screening algorithms. However, impacts on this preterm population, families, and their healthcare team remain highly discrepant as compared to full-term newborns.

Serial screening algorithms exist for low birth weight, preterm, and sick newborns, and are utilized by many NBS programs to reduce unnecessary confirmatory testing. Some of these algorithms tend to end with a final specimen collected around one month of age if the child is still inpatient. For SCID/TCL screening, however, this may still not be enough time for TREC values to reach their normal physiological levels. To combat this, some programs have recommended continued biweekly or monthly screening until the newborn reaches 37 weeks corrected gestational age. This practice was implemented to build upon already existing serial screening procedures for premature newborns for other conditions, particularly congenital hypothyroidism, and has resulted in fewer unnecessary clinical confirmatory tests. However, while this extended serial screening process may eventually achieve resolution of the screen-positive results without involving clinical diagnostic testing, it also can result in a preterm newborn requiring far too numerous blood draws and screens over time. Indeed, approximately one in six LBW/premature newborns in this study required at least five newborn screens, even though few actually had SCID or TCL. 

## 5. Recommendations for SCID Follow-Up in the LBW/Premature Population

As per previous reports [12], TREC values were lower in preterm/low birth weight newborns and steadily increased as the newborn reached 37 weeks corrected gestational age. The likelihood of screening results finally normalizing increases by 11 percentage points after 36 weeks gestational age. These data suggest that programs may be able to strike a balance between ensuring early detection and not unnecessarily overburdening preterm families or their healthcare providers.

Based on the data above and the trajectory of TREC values over time, a recommendation for consideration is for programs to implement the standard serial screening protocol as suggested by CLSI NBS03 [13], with assurance that a screen is collected after the child reaches at least 32 weeks corrected gestational age, at which point TREC results begin to normalize. For some newborns, the routine serial screening protocol will suffice to meet this timeframe, but for very or extremely premature infants this may necessitate another screen at this time. Regardless, if TREC values still have not normalized on this screen, another repeat screen should be considered at 37 weeks corrected gestational age, or at discharge, whichever is earliest. With any serial screening model utilized in the premature population, it is important to still utilize an urgent cutoff value (usually no TRECs) and appropriately triage these newborns for evaluation, rather than wait for another NBS sample. A model such as this takes into account the delayed normalization of TREC values in this population, and can allow for quick detection while minimizing extra blood draws and limiting provider and family burden.

The findings in this report are subject to limitations. Seven states participated in this study, representing approximately only twenty-seven percent of total births in the United States. Additionally, due to the wide variations across both laboratory and follow-up practices for SCID screening, our project team was presented with several considerations when trying to aggregate data across states:Data collection/linkage: There were variabilities in which data the newborn screening programs collect (i.e., some programs do not collect gestational age), and whether newborn screening programs were able to link multiple specimens to a single patient or link diagnostic data to a screening result.Terminologies used: There is variation in the terminology used to report results (positive, abnormal, out-of-range, borderline). To mitigate this issue, the study team created uniform definitions for the purpose of the study and asked participants to utilize the study definitions regardless of the programmatic terminology employed internally. A single study team member then reviewed and audited the data provided, to ensure the standard use of the study definitions.Follow-up recommendations: There are also differing follow-up recommendations/courses of action after a positive result (i.e., waiting for routine repeat requests, asking for additional repeat specimens outside of the routine process, or sending newborns straight for clinical/diagnostic evaluation).

## 6. Conclusions

While NBS for SCID successfully identifies infants with both true/leaky SCID and TCL, the efficient completion of NBS for SCID in the premature/LBW population continues to present challenges. However, the aggregation of NBS data across a range of NBS programs has highlighted the trajectory of TREC values over time, both between and within newborns, and provides evidence for improved SCID screening recommendations in this population.

In addition to the work accomplished here for SCID NBS, the methods utilized for data collection (i.e., providing clear data definitions), data auditing, and data aggregation could be easily translatable to other NBS diseases across disparate NBS programs.

## Figures and Tables

**Figure 1 IJNS-09-00068-f001:**
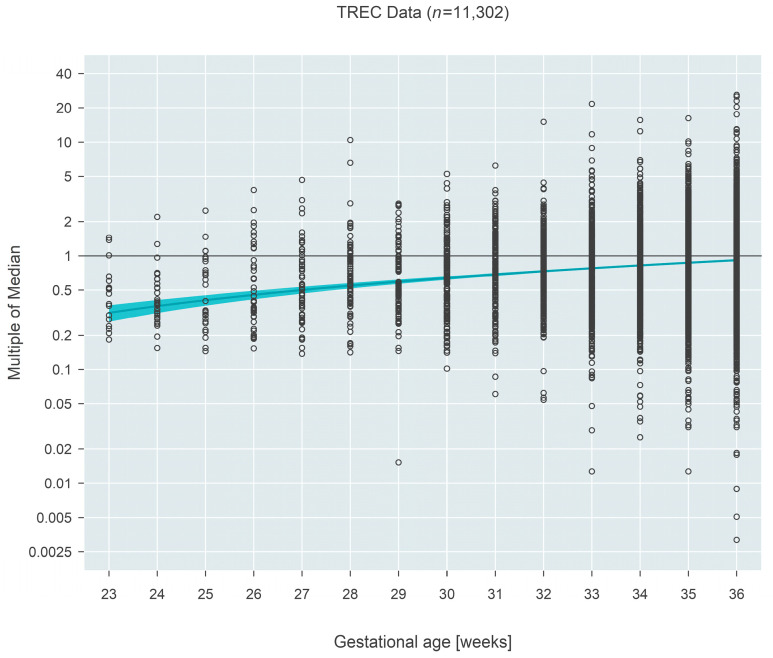
Multiple of the median (MoM) computed from TREC values from three states. The blue line and surrounding shaded 95% confidence region estimate how the median MoM varies as a function of gestational age. Between 23 and 36 weeks, the median MoM increases 2.9-fold from 0.32 to 0.92.

**Figure 2 IJNS-09-00068-f002:**
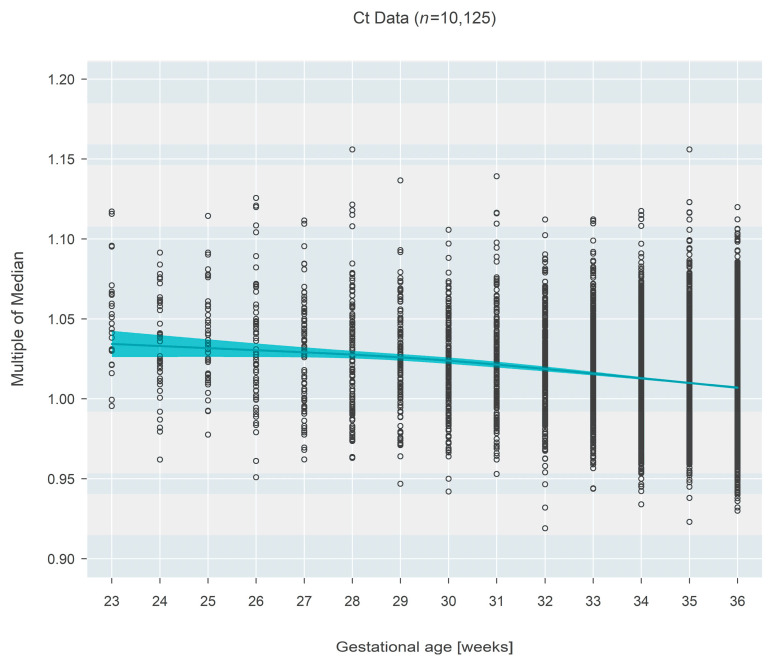
Multiple of the median (MoM) computed from Ct values from three states. The blue line and surrounding shaded 95% confidence region estimate how the median MoM varies as a function of gestational age. Between 23 and 36 weeks, the median MoM decreases by 2.6% from 1.034 to 1.007.

**Figure 3 IJNS-09-00068-f003:**
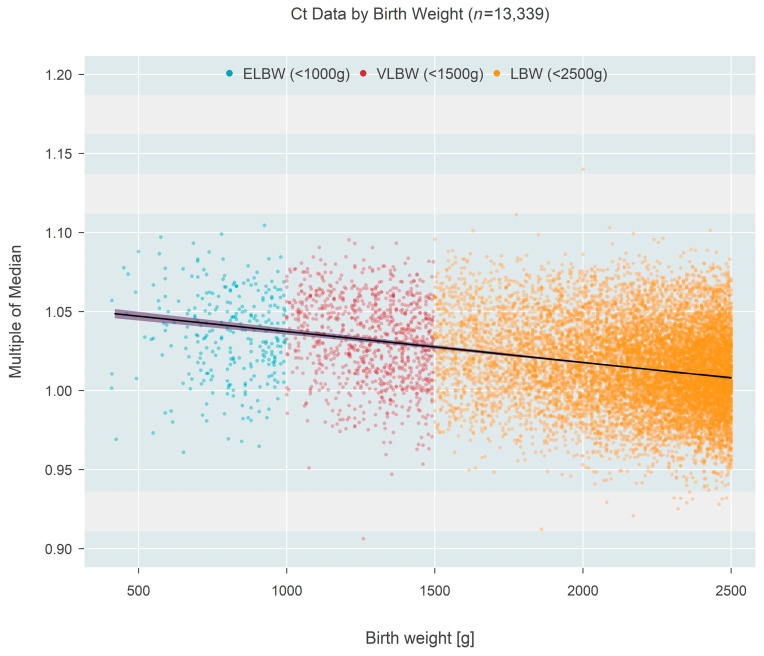
Multiple of the median (MoM) computed from Ct values of one state that recorded birth weight instead of gestational age. Individual points are colored according to birth weight categories. The black line and surrounding shaded 95% confidence region estimate how the median MoM varies as a function of birth weight. A 500 g increase in birth weight is associated with a 0.96% reduction in the median MoM.

**Figure 4 IJNS-09-00068-f004:**
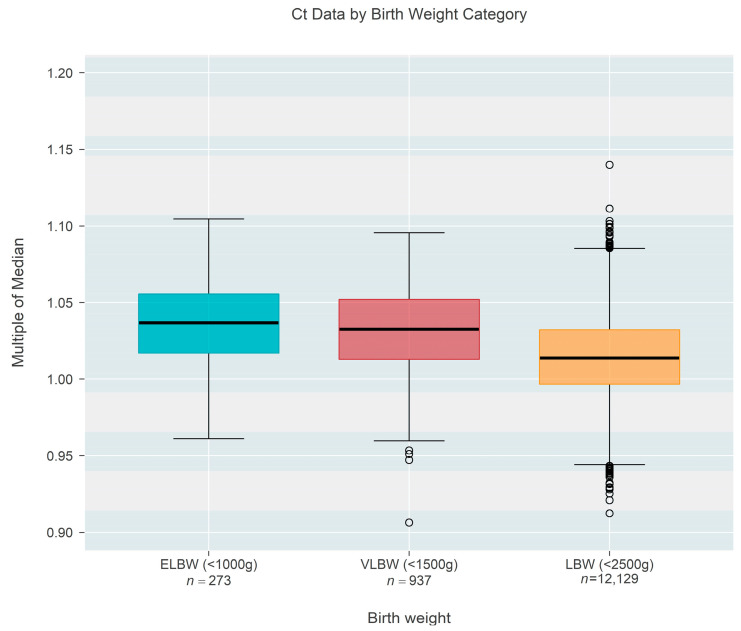
Box-whisker plots of the multiple of the median (MoM) results computed from Ct values of one state that recorded birth weight instead of gestational age. Upper and lower edges of each colored box mark the 75th and 25th percentiles, respectively, and display a span of one inter-quartile range [IQR]. The central black line within each box is the median; whiskers extend from box edges to the smallest/largest observation that remains within 1.5 IQR of the box edge; outliers are shown as separate points. There is no meaningful difference between the two lowest categories of birth weight, but each of the lower two differs strongly from the last (largest) category.

**Figure 5 IJNS-09-00068-f005:**
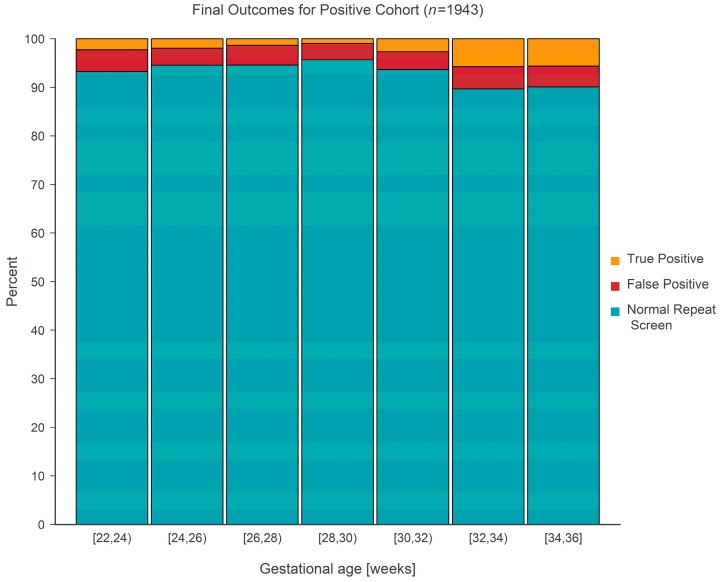
Stacked bar chart for the positive cohort showing the distribution of final outcome(s) according to gestational age.

**Figure 6 IJNS-09-00068-f006:**
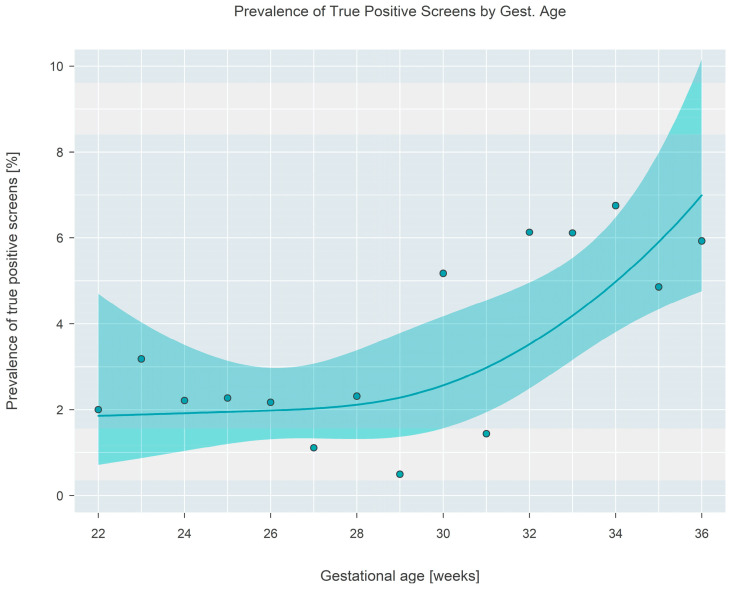
Prevalence of true positive final outcomes at each gestational age. A fitted solid line and surrounding shaded 95% confidence interval are shown, with filled circles showing the proportion observed in the sample at each gestational age.

**Figure 7 IJNS-09-00068-f007:**
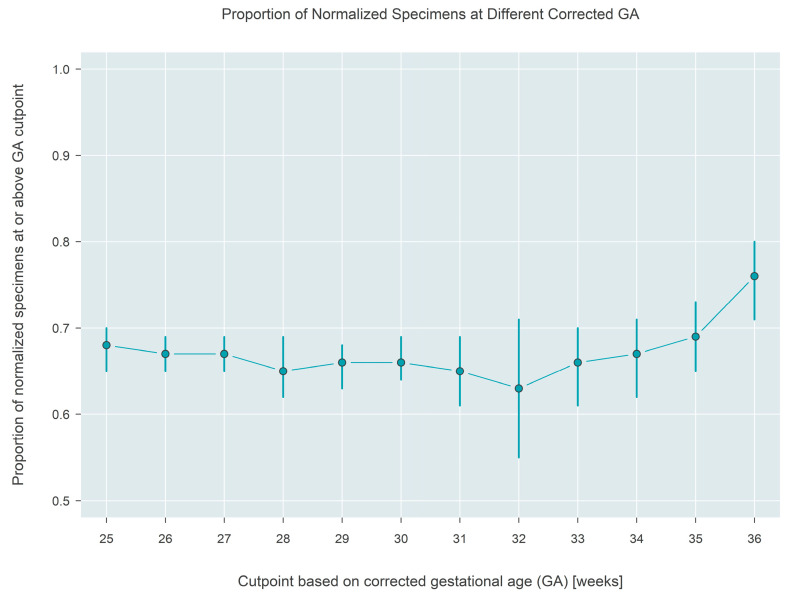
Estimated proportion of a normalized result from the testing algorithm applied to samples collected at different corrected gestational ages. Approximately 76% of samples collected at or after 36 weeks (corrected) gestational age are normalized.

**Table 1 IJNS-09-00068-t001:** Frequency and percentage (in parentheses) of final outcomes for the positive cohort according to gestational age.

Gest. Age (Weeks)	NRS ^1^	FP ^1^	TP ^1^
[22, 24)	124 (93.2)	6 (4.5)	3 (2.3)
[24, 26)	293 (94.5)	11 (3.5)	6 (1.9)
[26, 28)	278 (94.6)	12 (4.1)	4 (1.4)
[28, 30)	199 (95.7)	7 (3.4)	2 (1.0)
[30, 32)	177 (93.7)	7 (3.7)	5 (2.6)
[32, 34)	218 (89.7)	11 (4.5)	14 (5.8)
[34, 36]	510 (90.1)	24 (4.2)	32 (5.7)

^1^ NRS = normal repeat screen; FP = false positive; TP = true positive.

## Data Availability

The de-identified datasets presented in this article are not readily available, because access requires permission from state and territorial newborn screening programs. Requests to access the datasets should be directed to ruthanne.sheller@aphl.org.

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
