# Peer review of "Newborn Screening for Severe Combined Immunodeficiency: Lessons Learned from Screening and Follow-Up of the Preterm Newborn Population"

_2409-515X, 2023, doi:10.3390/ijns9040068_

Round 1

Reviewer 1 Report

Comments and Suggestions for Authors

Dear Authors:

I consider that the results provided by your work can be useful to improve SCID NBS. Furthermore, the manuscript is very well explained, written and includes all the relevant aspects that should be reflected in a scientific article. Below I leave some comments that I think can improve your work.

Introduction.

Line 36. Please, add “enzyme replacement therapy”.

Line 49. Please, rewrite the sentence to include other non-RT-PCR methodologies employed (for example Enlite Neonatal TREC kit used in California NBS).

Line 56-57. If it is described in the literature, could you indicate an approximate percentage of FP results for SCID NBS because a preterm newborns?

Materials and Methods

Line 118. Please, check if you mean “invalid” instead of “valid” (comparing with Supplementary data)

A)    General commentary: It is controversial to show TREC values evolution based on birth weight (BW) in preterm newborns. This is fine only as indirect measure of gestational weeks (GW), because in this case BW is a confounding variable. To study the BW influence, a sample of term newborn population should have been used. You have to clarify that when BW is used an indirect measure of GW is shown. In summary the authors have two options: skipping results of TREC in base on BW, or clarifying BW as indirect measure of GW in preterm newborns.

So you have to review and express all your result about this field with caution in this paper. The reader should not interpret TREC as birth weight-dependent based on the results observed in this work.

Results

Line 208.  Please, clarify if normal repeat screen is in the same first sample or in a new one (second sample).

Discussion

I agree with your recommendation (37 weeks) explained in line 280. However, according your results shown in figure 5 and 6, you should ask first an additional sample at >32 GW corrected, and then if not normalization of TREC values is observed, and ask for an additional one at 37 GW corrected (or at discharge, whichever is earliest). If the authors are not agree with this, please, justify to me. Thanks.  

Please, in line 281, specify that an alarm values have to be established by the NBS lab for preterm newborns. If you have a TREC value < urgent cutoff, you have to directly referral and don’t wait for a second sample.

Conclusions

Please, specify your specific recommendation for preterm newborns in SCID NBS.

Author Response

Thank you so much for your very thoughtful review! Please see the attachment for responses.

Reviewer 2 Report

Comments and Suggestions for Authors

The authors of the manuscript entitled "Newborn Screening for Severe Combined Immunodeficiency: Lessons Learned from Screening and Follow-Up of the Preterm Newborn Population" have confirmed data from previous publications emphasizing the need for adjusted screening algorithms for preterm infants in NBS for SCID. Their unique approach of using MoM and their sound statistical analysis provide an new approach to analyze divergent screening data and parameters across different states or countries.

The manuscript is well written and the data is presented in elaborate plots and graphs. I have some minor remarks.

Line 119: ‘Newborns with screen-positive results for disorders not classified as SCID/TCL were not included’. Which disorders have the authors excluded and why?

Line 157. The methods state that you have collected both gestational age and birth weight as parameters in your cohort (also shown in the Supplemental data). Line 157 implies that of the states only shared birth weight and not matching gestational age, is this correct? Other states did not also share data on birth weight that could be incorporated into figure 3a/b? 

Line 208. Maybe highlight that this data refers to the Aggregate Data Set the authors mention in the M&M.

Line 211. What do the authors define here as transient T-cell lymphopenia? Newborns with normal T-cell subsets at clinical evaluation or newborns that had low T-cell numbers that normalized over time in follow-up? Do authors for example also include congenital heart defects in this group with T-cell numbers that normalized after heart surgery? Please specify.

Discussion line. 278-284. I completely agree with the recommendations of authors to implement one additional screen once the child reaches 37weeks corrected gestational age, however can the authors comment on the counter arguments of some countries stating they might miss SCID/TCL cases in preterm infants?  

Discussion line 294. Terminology, this remains an important issue when comparing screening data in NBS programs around the globe (https://pubmed.ncbi.nlm.nih.gov/34537207/) How did the authors overcome these challenges (including data collection/linkage and follow-up recommendations)/

Conclusion. The authors provide a very generic conclusion, but I would definitely include their unique methodological and statistical approaches to compare and combine different screening data across states. These type of methods could also be used for other screening outcomes and NBS programs for other disorders.

Author Response

Thank you so much for your very thoughtful review. Please see the attachment for responses.
